# Do Gamers Play for Money? A Moderated Mediation of Gaming Motives, Relative Deprivation, and Upward Mobility

**DOI:** 10.3390/ijerph192215384

**Published:** 2022-11-21

**Authors:** Anthony King, Gloria Wong-Padoongpatt

**Affiliations:** Department of Psychology, University of Nevada, Las Vegas, Las Vegas, NV 89154, USA

**Keywords:** internet gaming disorder, relative deprivation, video game, gambling, upward mobility

## Abstract

Past research indicates strong monetary motives for gambling often elevate an individual’s risk of experiencing symptoms of gambling disorder, with personal relative deprivation (PRD) and upward mobility (UM) identified as key factors in this relationship. Nevertheless, few studies have examined how financial motives, PRD, and UM might interact for people playing modern video games—many of which offer financial incentives to encourage participation. Due to the overlap between gambling and (video) gaming, evidence suggests disordered gambling and disordered gaming might also share similarities. Therefore, the present study explored whether PRD influences associations between playing video games for financial motives, symptoms of Internet gaming disorder (IGD), and UM in two samples: 797 college students (Study 1) and 179 adult gamers over 25 years old (Study 2). Results from Study 1 revealed more PRD predicted more IGD symptoms, with higher financial gaming motives mediating the relationship. In Study 2, PRD also predicted IGD severity, but only coping motives appeared to mediate the positive association between PRD and IGD severity. In both samples, perceived UM inversely moderated the effect of PRD on one’s financial or coping gaming motives. These findings suggest financial motives for video games might lead to more problematic forms of participation for younger adults and negative perceptions of PRD and UM might interact, similar to gambling, to elevate a gamer’s vulnerability for IGD.

## 1. Introduction

Nowadays, video gaming and gambling are considered socially acceptable behaviors. In fact, not only are they acceptable, these pastimes are immensely popular. As a result, access to these forms of entertainment has skyrocketed in recent decades, often leaving policymakers and public health officials playing catch-up with the rapid advancements of these industries [1,2]. One major advancement has been that many popular video games are currently offering financial incentives and rewards to drive consumer business within online gaming platforms (e.g., online game Axie Infinity). In other cases, video games may not offer direct monetary rewards to gamers, but instead, gamers might bet or compete for money, sometimes millions of dollars (e.g., Fortnite World Cup 2019), through official esports tournaments or unofficial, private game matches. Despite these recent gaming trends, though, little is known about how gamers’ financial motives are related to disordered gaming.

In past samples of gamblers, a consistent pattern has emerged: stronger financial gambling motives are associated with greater gambling frequencies and gambling-related problems [3]. An interaction has also been found between higher levels of perceived personal relative deprivation (PRD) and lower levels of upward mobility (UM) that magnifies this effect for some gamblers [4,5]. It is possible, based upon the increasing convergence of gaming and gambling, that some gamers might follow a similar trajectory through high PRD and low UM towards gaming-related problems with financial motives for modern video games [6,7,8,9,10]. Therefore, the primary objective of this research was to explore the mediating role of financial gaming motives in the associations between PRD and symptoms of Internet gaming disorder (IGD; American Psychiatric Association [APA, [11]]), in addition to investigating the moderating effect of UM on the PRD-financial-motives pathway. The secondary objective was to establish baseline rates for these variables in both young adult gamers (Study 1) and older adult gamers (Study 2) from the United States (U.S.), since limited data exist.

### 1.1. Gaming and Gambling Overlap

Video gaming has been linked to recreational and problematic gambling for over three decades now [12,13,14,15,16,17,18,19,20,21]. Fisher and Griffiths [22] examined this topic in the mid-1990s and found that both video games and slot machines of that era shared the following characteristics: (1) a gaming loop controlled the players responses to stimuli, (2) players were usually required to have good hand–eye coordination, (3) rapid gameplay was at least partially dependent on a player’s skill level, (4) game rewards were accompanied by audio and visual effects, (5) rewards (e.g., points, trophies, money) increased when a player was winning, (6) accumulated rewards were visually displayed, and (7) there was an opportunity for receiving positive recognition from one’s peers based on the game’s competition. Critically, all of those features are still present in today’s video games and gambling machines.

This raises the question: if this gaming–gambling relationship has been suggested for multiple decades, and the core features of these games are essentially the same as their predecessors, why is it that the APA and the World Health Organization (WHO) have only recently started to consider IGD as a potential medical diagnosis? Pathological gambling, after all, has been recognized as a mental disorder for more than four decades [23,24]. What has changed in the world of video games and in the lives of problematic gamers to prompt the sudden re-evaluation of this behavior as a potential addictive disorder?

### 1.2. Making Real Money Playing Video Games

One significant change that has occurred in the video gaming world is the central role that money now plays. Even though the fundamentals of today’s electronic games have remained relatively unaltered over the years, the ways in which these design features are currently monetized have been reinvented completely. Depending on the specific video game title being played, virtual items obtained from modern in-game purchases (i.e., microtransactions), as well as other aspects of gameplay, can have a real monetary value linked to them. Typically, the rarer an item is in the virtual world, the more financial value it holds in the marketplace. For many video games, virtual items can be bought, sold, traded, and even wagered amongst players of all ages [25]. These activities can take place either through the developer’s authorized website, such as Steam, or an unauthorized, third-party platform, such as Skinwallet [26,27].

Beyond money-making virtual items in modern gaming, though, it is also possible now to make life-changing money from simply playing video games. Despite this being a reality that 99.9% of players never see (i.e., <500 professional gamers [28]), the recent rise in the popularity of esports, or competitive video gaming, has made playing video games a serious money-maker for talented, as well as lucky, players [29,30]. In 2019, for example, 16-year-old Kyle Gierdorf won the Fortnite World Cup tournament, earning himself a payday of roughly USD 3 million [31]. Stories like this one have made esports one of the fastest growing spectator sports in the world, opening up an entirely new and untapped gaming–gambling platform with prodigious viewership potential [32]. Although young Kyle’s windfall may be inspirational in nature for many gamers—akin to the hope gamblers experience seeing a lottery winner appear on television—these unlikely outcomes might also create cognitive distortions in some vulnerable users that could lead to excessive, repetitive engagement with potentially serious ramifications.

Interestingly, both the APA and the WHO have yet to directly acknowledge the financial motives and risks that can be involved in problem video gaming. In fact, the APA’s Diagnostic and Statistical Manual of Mental Disorders-Fifth Edition (DSM-5 [11]) even states that their proposed IGD is considered separate from gambling disorder specifically “…because money is not at risk” (p. 797). Nevertheless, recent research on this topic continues to suggest that financial motives and risks need to be considered when discussing the potential harms associated with gaming disorders [33,34,35,36,37] and video game purchase options, such as loot boxes, that rely on gambling mechanics [38,39,40,41,42].

Close et al. [43] analyzed data from 7767 video gamers who had previously made in-game purchases and found that the highest 5% of spenders accounted for approximately half of the in-game revenue generated. Perhaps coincidentally, this trend mirrors the pattern that has been observed in the casino business for decades [44], with a small fraction of high-rollers allowing the industry to thrive. Crucially, though, these top spenders for video games (>USD 100/month) were not high-income earning individuals, and around one-third of them were classified as problem gamblers. Thus, it appears that a disproportionate amount of current video game profits is derived from players who are more likely to report symptoms of gaming and/or gambling disorders [43].

### 1.3. PRD, UM, Money Motives, and Gambling Disorder—What about Gaming?

One common factor that could elevate someone’s vulnerability for gaming and gambling disorders is called *relative deprivation*: the perception that, in comparison to similar others, one has less than they rightly deserve—often due to circumstances beyond their control [45]. In some instances, this frustration or distress can be perceived at a group level and lead to collective social action; whereas, in other circumstances, the effects of relative deprivation may only be noticeable at an individual level (i.e., PRD) and may lead to a wide range of different coping responses [46]. However, more often than not, PRD appears to elevate risky coping behaviors, such as heavy alcohol use, opposed to coping strategies with less potential for harmful consequences, such as cardiovascular exercise [4]. Several studies have also linked more PRD to poorer mental and physical health outcomes, in part, as a result of using maladaptive coping behaviors over a long period of time (for review, see [46]).

Nevertheless, there are certain contexts in which PRD might lead to healthier coping strategies [47]. Such PRD scenarios typically offer multiple ways that a person can reasonably reduce their feelings of PRD (e.g., getting additional employment or education) without taking extraordinary risks to achieve some referent standard. Thus, it appears that when opportunities for upward mobility (UM)—defined as the capacity to rise to a higher socioeconomic status [SES])—are abundant and widely accessible, PRD might quite possibly be the encouragement some people need to improve the quality of their lives, but unfortunately, it seems many of the previous conventional pathways that tend to increase UM have largely disappeared or become much harder to achieve for the average person [48,49]. In places such as the U.S., PRD is especially prevalent, with around 60% of adults perceiving substantial inequalities in their lives that will likely require major overhauls to the current systems in order to correct [50]. With conventional, or less risky, options for UM unavailable, current evidence suggests that some people experiencing PRD will gravitate towards more unconventional, or riskier, means to correct their perceived situations [45].

Bernburg et al. [51] found that for some adolescents experiencing PRD, they would attempt to improve their comparative status with similar others by committing violent acts or engaging in other deviant behaviors (e.g., stealing), with the highest rates occurring in schools that were considered the least objectively deprived. For adults, on the other hand, one of the most common, yet still unconventional ways some people attempt to rectify PRD is through gambling [52,53,54], and several studies have shown that strong financial motives for gambling are key indicators of disordered gambling [55,56,57,58]. Since higher levels of economic deprivation, whether subjective or objective, can enhance one’s motives for money-making activities, PRD has been implicated as a major contributor to problematic gambling [5].

Before the emergence of online video games, gambling was one of the few activities available to most people that offered a quick solution—or the mere hope of one—to correcting a perceived imbalance in a person’s life. In reality, however, advancements in online technology have allowed these once separate game types to blur beyond distinction (e.g., social casino games). Therefore, considering that 16-year-olds are becoming millionaires from playing video games and virtual items from many of these games are serving as surrogate currencies [26], it is possible that gamers who perceive more PRD and less UM might view video gaming as a viable way to improve their financial status.

In total, one previous study has investigated the relationship between PRD and IGD. Qian et al. [59] found that financial PRD and symptoms of online gaming addiction were positively associated in a sample of Chinese college students (*n* = 1008). Besides the abstract, however, Qian et al.’s article was written in a non-English language, which restricted the details that were obtained from the study. Based on this limited evidence, it is unclear whether PRD and IGD share a similar association in U.S. college students, and how this association might further relate to different video game motives and perceptions of UM. Additionally, it is still unknown if PRD is relevant to older adult gamers, and how it might be connected to their IGD symptoms and gaming motives as well.

### 1.4. Current Study

Therefore, the present research sought to address these gaps in the literature and examine the impact of different gaming motives and UM perceptions on the association between PRD and IGD severity in U.S. college students (Study 1). We also explored if these associations would appear in adult gamers over the age of 25 from the U.S. population (Study 2). In both studies, we used a mediation model to initially test the connection between PRD and IGD severity via financial video game motives. We hypothesized PRD would have a positive association with IGD severity (H1), and financial gaming motives would mediate this relationship (H2). Next, a moderated mediation model investigated whether the indirect association between PRD and IGD severity via financial gaming motives was moderated by perceived UM (see Figure 1 for a conceptual diagram). Specifically, this model examined if perceived UM moderated the association between PRD and financial gaming motives. We hypothesized that as perceived UM decreased, gamers with higher levels of PRD would have more financial motives for video games (H3) and in turn, more symptoms of IGD (H4). While the primary hypotheses of this research focused on financial gaming motives, we included any other gaming motives (i.e., emotional coping, mood enhancement, and social motives) in our models if they were significantly associated with both PRD and IGD severity.

## 2. Materials and Methods

### 2.1. Participants and Procedure—Study 1

College students (*n* = 797) were recruited to participate in an online, cross-sectional survey via an undergraduate psychological research participant pool. This survey was made available to the university’s students if they met the following eligibility requirements: at least 18 years old, living in the U.S., and fluent in the English language. Eligible participants self-selected to take the survey and were compensated with one course unit of participation credit. To avoid any prior indication of the survey’s focus on video game behaviors, both gamers and non-gamers were able to participate. Nevertheless, participants who indicated they had not played video games two or more times during the past 12 months were excluded from the final analytic sample. Out of 1048 college student respondents, 797 (76%) were eligible for inclusion in the final dataset based on their video gaming frequency and data quality (i.e., attention checks passed and survey time > 7 min). The mean age of the sample was 20.30 years (*SD* = 4.54), with 62.9% of participants identifying as female (see Table 1 for additional demographics). The Institutional Review Board at the University of Nevada, Las Vegas (UNLV) approved this research as exempt, and all participants provided electronic informed consent prior to their participation in the survey. Data for Study 1 were collected from 1 September 2020 to 28 April 2021.

### 2.2. Participants and Procedure—Study 2

Older adults (*n* = 179) from the U.S. were recruited via the Amazon Mechanical Turk (MTurk) website. The MTurk platform is a crowdsourcing marketplace that allows people to complete online assignments for money. Previous research has indicated MTurk as a reliable and valid source for data related to different addictions [54]. The same survey from Study 1 was used in Study 2 and was available to any MTurk Masters, which are workers with excellent performance ratings, who were: at least 25 years old, living in the U.S., and fluent in the English language. Eligible participants self-selected to participate in the study and were compensated USD 2.00 for completing the survey. As in Study 1, to avoid indications of the survey’s topic, both gamers and non-gamers were allowed to participate, with the same gaming frequency and data quality requirements for inclusion in the analytic sample. Out of 228 respondents, 179 (78.5%) were eligible for the final dataset. The mean age of the sample was 41.87 years (*SD* = 10.16), with 48.6% of participants identifying as female (see Table 1 for additional demographics). The Institutional Review Board at UNLV approved this research as exempt, and all participants provided electronic informed consent prior to their participation in the survey. Data for Study 2 were collected from 5 April to 13 May 2021.

### 2.3. Measures

Composite scores for each measure were calculated by averaging responses to scale items (except for IGD measure). Higher scores reflect more of the measured construct. See Table 2 and Table 3 for bivariate correlations, descriptive statistics, and internal consistencies of the measured variables in both studies.

#### 2.3.1. Personal Relative Deprivation (PRD)

PRD was measured using Callen et al.’s [60] 5-item questionnaire (e.g., “When I think about what I have compared to others, I feel deprived”). Responses to each item were anchored at *strongly disagree* (−3) to *strongly agree* (3), with items 2 and 4 reverse-coded. All questions on this measure were asked in reference to three levels: (1) the national level (i.e., the entire US), (2) the community level (e.g., neighborhood or city) and (3) the online community level (e.g., social media or other internet-based communities). Responses for each level were combined to create a single score. This questionnaire has previously demonstrated adequate internal consistency (α = 0.78), in addition to good construct and criterion validity in predicting consequences related to PRD (e.g., the desire for immediate rewards) in samples of gamblers from the general population [60].

#### 2.3.2. Perceived Upward Mobility (UM)

Perceived UM was assessed using a 2-item questionnaire [5]. The items on this questionnaire include: “In this country, I will be able to find a job that will improve my financial situation,” and “In this country, if I work hard enough, I will be able to improve my financial situation.” Responses to these items ranged from *strongly disagree* (−3) to *strongly agree* (3). This scale has shown acceptable construct validity in measuring economic UM in U.S. gamblers, and good criterion validity in predicting the association between PRD and financial gambling motives [5]. Sufficient internal consistency for these two questions was demonstrated in both of our samples.

#### 2.3.3. Internet Gaming Disorder (IGD)

Disordered gaming behaviors were assessed using the IGD Scale (IGDS-9 [61]), which is a 9-item measure based upon the proposed IGD diagnostic criteria of the DSM-5 [11]. For example, “During the last 12 months, have you been feeling miserable when you were unable to play a [video] game?” Each item of the IGDS-9 used a dichotomous response option: *no* (0) and *yes* (1). Items were summed to indicate IGD severity. This scale has displayed strong construct, criterion, and structural validity for measuring IGD symptoms and its associations with other related behaviors in large representative samples of adolescents and adults [61]. Internal consistency for this scale was satisfactory in our samples.

#### 2.3.4. Video Game Motives

Video game motives were measured using a modified version of the Gambling Motives Questionnaire-Financial (GMQ-F [62]). This measure examines motives within four domains: emotional coping, mood enhancement, social, and financial. Originally, the GMQ [63] was adapted from the Drinking Motives Questionnaire [64] and did not include a full financial motives subscale until Dechant et al.’s [62] version. Since the first GMQ scale was adapted from a measure about drinking alcohol because their motives appeared similar, we applied this reasoning to gambling and video gaming. As shown in earlier studies, video gamers and gamblers share several characteristics and motives, such as elevated risk taking and playing to socialize [65,66].

This scale contained 15 items (e.g., “How often do you play video games to be sociable?”) scored 1 (*never or almost never*) to 4 (*almost always or always*); one financial motive item from GMQ-F was removed because the language did not easily modify to video games (i.e., “How often do you gamble because you enjoy thinking about what you would do if you won a jackpot?”). In each question, the only modification was changing “gamble” to “play video games.” Scores were summed for each motive independently. We also performed a confirmatory factor analysis on this modified scale for both studies, which indicated this modification of GMQ-F had adequate psychometric properties. The GMQ-F has demonstrated sufficient construct and statistical validity, as well as internal reliability, in measuring distinct motives associated with gambling behaviors in large representative samples from Canada [67], France [68], and the U.S. [5].

#### 2.3.5. Covariates and Demographics

Impulsivity was measured using the Short UPPS-P Impulsive Behavior Scale [69]. This scale includes a total of 20 items (e.g., “I tend to lose control when I am in a great mood”), with four choice options ranging from *strongly agree* (1) to *strongly disagree* (4). This instrument measures impulsivity related to five distinct domains: sensation seeking, lack of premeditation, lack of perseverance, negative urgency, and positive urgency. Prior to the DSM-5 [11], gambling disorder was classified as an impulse control disorder for multiple decades, instead of as an addictive disorder. Despite this recent reclassification, poor impulse control remains a common risk factor for an array of addictions [70,71]. Thus, to ensure our analytic models are driven by PRD, we controlled for gamers’ impulsivity. In non-clinical adult samples from France [72], Italy [73], and the U.S. [74], this scale has demonstrated acceptable internal consistency (α = 0.70–0.85), in addition to displaying adequate construct and criterion validity in distinguishing different forms of impulsivity and their related behaviors, such as problem gambling and drug use.

Loneliness was measured using the 6-item Revised UCLA Loneliness Scale (RULS-6 [75]). Each of the items (e.g., “How often do you feel alone?”) are scored from 1 (*never*) to 4 (*always*). Similar to impulsive behaviors, previous research has linked loneliness to problematic gambling and gaming [76,77]. Therefore, loneliness was controlled for, in order to avoid possible third-variable explanations for our results. This scale has displayed adequate convergent and discriminant validity, as well as sufficient internal consistency (α = 0.72–0.84), in both student and clinical populations in Thailand [75]. Thus, the RULS-6 appears to provide a briefer alternative to one of the most widely used measures of loneliness in the adult population.

Furthermore, gender and personal income were measured as potential covariates for our analyses. Previous research on gambling [78] and video gaming [36] has indicated males as being at an elevated risk for problematic involvement in these game-related behaviors. Additionally, lower levels of personal income have been linked to more PRD and financial motives for gambling [5], which could be applicable to financial video game motives as well. Demographic information was also collected related to the age and race/ethnicity of participants, but these data were not controlled for due to less theoretical justification for their inclusion, unrepresentative characteristics of the samples, and our reliance on convenience sampling. Personal yearly income was reported using a multiple-choice question with 11 ordinal choice options: *less than USD 20,000* (1) to *USD 200,000 or more* (11).

### 2.4. Data Analysis Plan—Study 1 and 2

In accordance with a general analytical framework for conditional process analysis, mediation (PROCESS Model 4) and moderated mediation models (PROCESS Model 7) were conducted in SPSS v27 with PROCESS 3.4 macro [79,80]. These models first examined if PRD (independent variable) predicted IGD severity (dependent variable) indirectly through financial video game motives (mediator); the other three gaming motives were also evaluated as potential mediators if they demonstrated significant bivariate associations with our primary variables. The relationship between PRD and any significant mediators were then tested for interactions at low (−1 *SD* below *M*), average (*M*), and high (+1 *SD* above *M*) levels of perceived UM (moderator). To examine any moderated mediation effect, our final analysis investigated the conditional indirect effect of PRD on IGD severity via gaming motives at each level of perceived UM. All analyses included covariates displaying significant bivariate correlations to both PRD and IGD severity, or if they had a strong theoretical basis in the models, as was the case for personal income. Furthermore, due to the statistical literature currently lacking clear methods for calculating power in complex moderated mediation models [81], sample sizes in other related studies (e.g., [38]) were used as a reference point for adequate numbers of participants. Statistical significance was set at an alpha of 0.05 and determined by 95% confidence intervals that did not include zero (based on 5000 bootstrapped samples).

## 3. Results

### 3.1. Video Game Frequencies—Study 1

#### 3.1.1. Time and Money Spent and Gaming Devices Used

For college students, the average time spent playing video games was 2.29 h per day (*SD* = 1.99), and the average amount of money spent on in-game purchases during the previous month was USD 40.56 (*SD* = USD 59.10, Max = USD 500). Participants also estimated the total amount of money they had spent previously on a single video game’s in-game purchases (*M* = USD 158.64, *SD* = USD 371.96, Max = USD 3000). To facilitate their gameplay, participants reported using the following electronic gaming devices (multiple devices could be selected): 66.6% (*n* = 531) used a gaming console, 61.9% (*n* = 493) used a mobile device, and 42.8% (*n* = 341) used a personal computer (PC). PCs were the only device significantly associated with IGD severity (*r* = 0.29, *p* < 0.01).

#### 3.1.2. Internet Gaming Disorder Prevalence

Seventy-three participants (9.2%) reported five or more symptoms of IGD, which is currently the proposed threshold for diagnosis; notably, this group had a perfectly even gender distribution, with 36 females, 36 males, and one non-binary individual (*M*_age_ = 19.51, *SD* = 2.17). For these participants, several racial and ethnic backgrounds were reported: 43.8% (*n* = 32) were Asian American, 9.6% (*n* = 7) were African American/Black, 24.7% (*n* = 18) were Hispanic or Latin American, 32.9% (*n* = 24) were Caucasian American/White, 15.1% (*n* = 11) were Hawaiian/Pacific Islander, and 2.8% (*n* = 2) were American Indian/Alaskan Native and Middle Eastern. An additional 143 participants (17.9%) reported three to four IGD symptoms and could be considered a subclinical group for IGD diagnosis. See Table 1 for additional frequencies related to gameplay and money spent on in-game purchases.

#### 3.1.3. Financial Risks and Motives

In reference to the past 12 months, 56 participants (7.0% of entire sample or 24.7% of recent microtransaction spenders) reported experiencing problems from the money they spent on in-game purchases for video games, and 58 participants (7.3% of the entire sample or 25.6% of recent microtransaction spenders) reported hiding their in-game spending from others. Furthermore, 80 participants (10.1%) reported playing video games to WIN money at the following frequencies: 7.7% (*n* = 61) *sometimes*, 1.3% (*n* = 10) *often*, and 1.1% (*n* = 9) *almost always or always*. Sixty-four participants (8.1%) also reported playing video games to EARN money at the following frequencies: 4.9% (*n* = 39) *sometimes,* 1.9% (*n* = 15) *often*, and 1.3% (*n* = 10) *almost always or always.*

### 3.2. Preliminary Analysis—Study 1

Bivariate correlations indicated mood enhancement and social gaming motives, as well as impulsivity related to a lack of premeditation, a lack of perseverance, and sensation seeking were not significantly associated with PRD or IGD severity. Therefore, these mediators and covariates were excluded from our final analyses, and we controlled for gender, impulsivity related to negative and positive urgency, income, and loneliness in our models. Additionally, bivariate correlational analysis indicated PRD and IGD severity had a significant positive association (see Table 2), supporting our first hypothesis.

### 3.3. Mediation–Study 1

Overall, the mediation model explained 32% of the variance in IGD severity, based on its association with PRD via financial and coping gaming motives, *F*(7789) = 52.91, *p* < 0.001. PRD predicted financial gaming motives (*B* = 0.06, 95% CI [0.03, 0.09]), and financial gaming motives predicted IGD severity (*B* = 0.58, 95% CI [0.28, 0.89]). Despite PRD not predicting coping motives in this sample, coping motives did strongly predict IGD severity (*B* = 1.07, 95% CI [0.93, 1.21]). Financial gaming motives had a significant indirect effect on the association between PRD and IGD severity (*B* = 0.04, 95% CI [0.01, 0.07]), which indicated financial gaming motives as a full mediator in this relationship. Specifically, higher levels of PRD were associated with more financial gaming motives and in turn, more IGD symptoms reported; this result supported our second hypothesis.

### 3.4. Moderated Mediation–Study 1

The moderated mediation model revealed significant conditional effects of low and average levels of perceived UM on the association between PRD and financial gaming motives (see Table 4 for coefficients and confidence intervals). This result suggests the positive association between PRD and financial gaming motives was strengthened when there was less perceived UM (see Figure 2), which supports our third hypothesis. Moreover, a significant moderated mediation effect was found in this model (Index = −0.02, *SE* = 0.01, 95% CI [−0.05, −0.001]). This finding indicates the indirect relationship between PRD and IGD severity via financial gaming motives was significantly different for participants at each level of perceived UM. In particular, more PRD alone was not predictive of IGD severity via financial gaming motives if a participant reported high UM. It was solely when participants reported more PRD coupled with low and average levels of UM that this conditional effect was associated with more severe forms of IGD via financial motives, supporting our fourth hypothesis.

This moderated mediation model also had several significant associations with our covariates. For the relationship between PRD and financial gaming motives, males were indicated as being more likely to report these motives, as compared to females (*B* = 0.06, 95% CI [0.03, 0.09]). Nevertheless, gender was not predictive of IGD severity (*p* = 0.07). Further, more financial gaming motives were linked to less loneliness (*B* = −0.08, 95% CI [−0.12, −0.05]) and more positive urgency (*B* = 0.11, 95% CI [0.06, 0.15]), but they were not significantly connected to personal income (*p* = 0.60), although less income did predict more symptoms of IGD (*B* = −0.16, 95% CI [−0.31, −0.01]). For the overall model, both negative urgency (*B* = 0.24, 95% CI [0.02, 0.46]) and positive urgency (*B* = 0.38, 95% CI [0.16, 0.60]) were strongly related to IGD severity, suggesting impulsivity related to these domains plays a critical role in problem gaming behaviors. In sum, this model explained 12.6% of the variance in IGD severity, based on its association with PRD and UM via financial gaming motives, *F*(7789) = 18.23, *p* < 0.001. Without coping gaming motives included as a mediator in this model, approximately 20% less variance in IGD severity was accounted for in this sample, which suggests coping motives are more predictive of current IGD symptoms.

### 3.5. Video Game Frequencies–Study 2

#### 3.5.1. Time and Money Spent and Gaming Devices Used

For MTurk participants, the average time spent playing video games was 1.85 h per day (*SD* = 1.65), and the average amount of money spent on in-game purchases in the previous month was USD 36.67 (*SD* = USD 66.78, Max = USD 400). Participants also estimated the total amount of money they had spent previously on a single video game’s in-game purchases (*M* = USD 99.41, *SD* = USD 225.31, Max = USD 1500). To facilitate their gameplay, participants reported using the following electronic gaming devices (multiple devices could be selected): 57% (*n* = 102) used a gaming console, 62.6% (*n* = 112) used a mobile device, and 60.3% (*n* = 108) used a PC. Gaming consoles were the only device significantly associated with IGD severity (*r* = 0.19, *p* < 0.05).

#### 3.5.2. Internet Gaming Disorder Prevalence

Moreover, nine participants (5%) reported five or more symptoms of IGD; similar to the university participants, this group had a near-even gender distribution, with four females and five males (*M*_age_ = 36.44, *SD* = 10.54). For these participants, 33.3% (*n* = 3) were Asian American and 66.7% (*n* = 6) were Caucasian American/White. An additional 18 participants (10%) reported three to four IGD symptoms and could be considered a subclinical group for IGD diagnosis. See Table 1 for additional frequencies related to gameplay and money spent on in-game purchases.

#### 3.5.3. Financial Risks and Motives

In reference to the past 12 months, five participants (2.8% of entire sample or 11.6% of recent microtransaction spenders) reported experiencing problems from the money they spent on in-game purchases for video games, and three participants (1.7% of entire sample or 7% of recent microtransaction spenders) reported hiding their in-game spending from others. Furthermore, 22 participants (12.3%) reported playing video games to WIN money at the following frequencies: 9.5% (*n* = 17) *sometimes*, 2.2% (*n* = 4) *often*, and 0.6% (*n* = 1) *almost always or always*. Nineteen participants (10.6%) also reported playing video games to EARN money at the following frequencies: 7.3% (*n* = 13) *sometimes* and 3.4% (*n* = 6) *often.* Compared to university participants, these money motives for gaming were slightly higher for MTurk participants.

### 3.6. Preliminary Analysis–Study 2

Due to the exploratory nature of this second study, we included the same covariates in our models as Study 1. In addition, bivariate correlational analysis indicated PRD and IGD severity had a significant positive association (see Table 3), supporting our first hypothesis. Despite this significant relationship, financial gaming motives were not significantly connected to PRD, and therefore, our other three hypotheses were unable to be examined in this sample. Coping gaming motives, however, were significantly associated with both PRD and IGD severity. Therefore, we investigated how coping motives potentially mediated this relationship.

### 3.7. Mediation–Study 2

In this mediation model, PRD predicted coping gaming motives (*B* = 0.11, 95% CI [0.003, 0.22]), and coping gaming motives predicted IGD severity (*B* = 0.64, 95% CI [0.33, 0.94]). Gender was a significant covariate for the association between PRD and coping motives, with females being more likely than males to feel relatively deprived and game for emotional coping (*B* = −0.16, 95% CI [−0.32, −0.02]). However, personal income and positive urgency were not significantly connected to either coping gaming motives or IGD severity. For the total effect model without coping motives, 19.7% of the variance in IGD severity was explained by PRD and the five covariates, *F*(6172) = 7.01, *p* < 0.001. Specifically, IGD severity had significant associations with both negative urgency (*B* = 0.72, 95% CI [0.23, 1.21]) and loneliness (*B* = 0.35, 95% CI [.02, 0.68]), indicating higher levels of these constructs were related to more symptoms of IGD.

### 3.8. Moderated Mediation–Study 2

The moderated mediation model revealed significant interactions and conditional effects of low and average levels of perceived UM on the association between PRD and coping gaming motives (see Table 5 for coefficients and confidence intervals). Additionally, a significant test of the highest order for unconditional interactions between PRD and perceived UM in predicting coping motives was found, *F*(1170) = 5.24, Δ*R*^2^ = 2.4%, *p* = 0.02. These results suggest the positive association between PRD and coping gaming motives was strengthened when there was less perceived UM (see Figure 3).

Moreover, a significant moderated mediation effect was found in this model (Index = −0.05, *SE* = 0.03, 95% CI [−0.12, −0.003]). This finding indicates the indirect relationship between PRD and IGD severity via coping gaming motives was significantly different for participants at each level of perceived UM. In particular, more PRD alone was not predictive of IGD severity via coping gaming motives if a participant reported high UM. It was solely when participants reported more PRD coupled with low and average levels of UM that this conditional effect was associated with more severe forms of IGD via coping motives. In Study 1, a similar effect was observed, but in college students, it was linked to financial gaming motives instead of coping motives. In sum, this model explained 27% of the variance in IGD severity, based on its association with PRD and UM via coping gaming motives, *F*(7171) = 9.02, *p* < 0.001.

### 3.9. Exploratory Analyses–Study 1 and 2

#### Impact of the COVID-19 Pandemic on Video Gaming

To examine whether the COVID-19 pandemic impacted the amount of time and money that participants spent on video games, we asked participants to rate both items separately in comparison to before the pandemic started. For Study 1, 131 participants (16.4%) reported spending less time playing video games during the pandemic, 223 participants (28.0%) reported spending about the same amount of time, and 443 participants (55.6%) reported spending more time. For Study 2, 21 participants (11.7%) reported spending less time playing video games, 61 participants (34.1%) reported spending about the same amount of time, and 97 participants (49.2%) reported spending more time playing video games.

Regarding the pandemic’s effect on the amount of money spent on video games, in Study 1, 280 participants (35.1%) reported spending less money on video games during the pandemic, 363 participants (45.5%) reported spending about the same amount of money, and 154 participants (19.3%) reported spending more money. In Study 2, 17 participants (9.5%) reported spending less money on video games during the pandemic, 133 participants (74.3%) reported spending about the same amount of money, and 29 participants (16.2%) reported spending more money. Based on these results, it appears that the COVID-19 pandemic in the U.S. elevated the amount of time spent on video games for approximately 50% of players and the amount of money spent for 16.2–19.3% of players.

## 4. Discussion

### 4.1. Do Video Games Involve Financial Risks and Motives?

The short answer: yes. Consistent with previous research on the monetary aspects of modern video games [33,34,35,36,37,38,39,40,41,42,43], our results demonstrated that not only are financial motivations involved in today’s video games, but many popular video games pose a significant financial risk to some gamers. Using two demographically different U.S. samples, we observed that out of the gamers who bought in-game purchases, 9.3–12.8% spent over USD 100 during the previous month, with players from both samples reporting upwards of USD 400 to USD 500 spent within that same timeframe. Additionally, we inquired how much total money participants had ever spent on in-game purchases for a single video game. In both samples, between 13.7–18.9% of gamers with recent microtransaction purchases spent USD 200 or more within a single video game and between 2–4.6% of players reportedly spent USD 1000 or more. For the participants who spent USD 1000 or more, it is worth mentioning that one-third indicated they play the video game titles League of Legends and Valorant, which suggests certain games likely pose more of a financial risk than others. To add to this issue, these were not wealthy individuals spending large amounts of money on in-game purchases, since most of these players reported yearly incomes totaling less than USD 20,000.

Yet, the amount a player spends is not necessarily indicative of that person experiencing problems from that behavior. Therefore, we asked participants directly if the money they spent on in-game purchases (within the last 12 months) had caused them problems. For college students (Study 1), roughly one out of every four spenders (24.7%) stated they experienced some difficulty from the loss of their money to video game microtransactions. Whereas for MTurk participants (Study 2), which was a much older demographic group (*M_age_* = 41.87, *SD* = 10.16 vs. *M_age_* = 20.30, *SD* = 4.54), the rate of participants’ problems related to in-game purchases equated to about one out of every nine spenders (11.6%) experiencing problems from their financial involvement in video games. These reported rates of financial problems could possibly explain why 7–25.6% of microtransaction purchasers reported concealing their video game spending from others.

When it came to money motives for playing video games, we found that 10.1–12.3% of participants played video games to WIN money and 8.1–10.7% played video games to EARN money. Interestingly, MTurk participants had slightly higher rates of financial motives for gaming, but only in the college student sample did financial motives significantly connect to more reported symptoms of IGD. This pattern is, perhaps, due to college students being primarily young, low-earning individuals that likely have fewer financial resources to spare than older adult participants. It is also reasonable to assume that college students, opposed to MTurk workers, have less experience with traditional forms of employment, and in turn, they might allow their financial motives for gaming to reach more problematic levels than an older individual who has maintained conventional employment for several years. 

Nevertheless, these results indicate multiple key points relevant to the current conceptualizations of gaming and gambling disorder. First, modern video games that allow players to spend thousands of dollars on in-game purchases do represent a financial risk to some video gamers. This finding encourages a thorough re-evaluation of the APA’s current distinction between gaming and gambling disorders specifying that video games have no significant potential for financial loss. Second, if these disorders are to remain distinguished from one another in the eyes of the medical communities, then disordered video gamers with heavy financial involvement in gaming could be the first official subtype of IGD. After all, it is still unclear how different types of microtransactions, such as fixed-reward or randomized-reward, are tied to the financial problems that a subset of video gamers is reporting. Third, in all the analyses conducted, financial motives for video gaming were more effective in predicting a participant’s IGD severity than all other motives besides emotional coping. This result for financial motives is consistent with previous research on disordered gamblers [5,55,56,57,58] and it occurred despite the current IGD diagnostic criteria [11] revolving almost exclusively around the coping-related symptoms that can arise from one’s video game involvement. Although excessive time investment is undoubtedly a major harm to consider in examinations of problematic gaming, the escalating monetary risks involved in modern video games also deserve attention and acknowledgment from medical organizations attempting to define the range of negative consequences that can result from this behavior.

### 4.2. How Do PRD and UM Factor into IGD?

For college students, higher scores of PRD were strongly connected to more financial gaming motives, with a significant conditional effect on this relationship occurring at lower levels of perceived UM. In contrast, for MTurk participants, more PRD was associated exclusively with greater emotional coping motives for video games, with low levels of perceived UM creating a conditional effect on coping motives that was approximately three times the size of the effect that occurred for financial motives in the college sample. This result could indicate the various ways in which feelings of PRD can impact U.S. adults: in college students, it might move them towards more financial gaming motives, and in older adults, it might encourage more motives for emotional coping. In both instances, these motives indirectly connect feelings of PRD to greater problems experienced from video gaming.

Notably, perceived UM alone did not predict the number of IGD symptoms reported in either sample. These perceptions, however, did demonstrate significant negative associations with loneliness, as well as positive and negative urgency regardless of the participant’s age. Based on our results, higher levels of perceived UM appear to act as a protective factor to assist adult gamers with avoiding problematic gaming motives and behaviors. For both college students and older adults, we found as PRD increased and perceived UM decreased, participants became more prone to report financial or coping gaming motives—both of which connected directly to more severe forms of IGD. This finding could have particular relevance to our general understanding of what contributes to problem gaming behaviors in different U.S. groups and encourage clinicians to explore clients’ perceptions of UM, and the potentially exacerbating effect it may have on someone’s involvement in video games or other similar behaviors such as gambling.

### 4.3. So, Are Gaming and Gambling Disorders Related?

The long answer: absolutely. Although through the lens of a biopsychosocial framework of addictive disorders, all addiction-like behaviors are considered to be seeking the same result: external relief from internal discomfort. Even without the scientific data to back it (see [82,83]), most people seem to understand, at least on some level, that a healthy, non-stressed person does not require an addictive coping style to assist in regulating their psychological and physiological states. Therefore addiction, regardless of its opportunistic expressions, appears to be most accurately viewed as an adaptation that arises from the inseparable interaction of a person’s biological, psychological, and social environments. In the short-term, one’s addiction adaptation can be quite successful in achieving the desired consequence of feeling subjectively better—even if for only a moment. Over the long-term, though, the serious impairments of any addicted pattern tend to accumulate and can lead to substantial interference within a person’s life. 

In terms of gaming and gambling disorders specifically, ignoring the connections between different addictive disorders could increase the likelihood of their co-occurrence or potential substitution for one another. This possibility is supported by results from an extensive systematic review of the addiction literature [84], which examined 83 studies with sample sizes all above 500 and estimated that one out of every two U.S. adults experienced addiction symptoms within the last year. Unfortunately, psychiatric comorbidities (e.g., anxiety, depression, ADHD) and co-occurring addictions tend to be the rule, rather than the exception, when it comes to addicted individuals [84]. Thus, with gaming and gambling at the moment reaching more people than ever before in human history via online and mobile technology [2], it is difficult to imagine a scenario where this nearly unavoidable access to these activities reduces the number of people negatively impacted by these industries and the behaviors they encourage.

### 4.4. Implications…

#### 4.4.1. … for Theory

Along with this study’s contributions to the theoretical conceptualizations of gaming and gambling disorders, to the best of our knowledge, this research was the first attempt to move PRD theory beyond the real world and into the context of online communities. In general, we saw a trend with younger participants who specified they were active in online communities appear more susceptible to reporting higher levels of PRD. This finding should prompt the general public—particularly users of social media—to critically evaluate the potential societal effects that can arise from online social interactions increasingly substituting for real-world ones. Unlike social comparisons conducted in-person, online comparisons offer a global reference group and can obscure details that are needed for accurate evaluations of others, as well as of ourselves. When this point is considered within the evolutionary setting that our brains developed (i.e., relatively small, tribal groups), it seems reasonable that the human mind is perhaps simply unprepared to make thousands, if not millions, of social comparisons online without repercussions.

#### 4.4.2. … for Practice 

Furthermore, this research has important clinical implications for medical professionals treating addictive disorders. If our results are a reasonable reflection of the overall IGD rates in the U.S. general population (i.e., 5–9.2%), clinicians should be encouraged to target adult clients’ beliefs related to PRD and examine how these beliefs can maintain someone’s financial and emotional coping motives for video gaming. Since both subjective and objective data suggest the U.S. has substantially less UM than many other developed nations (i.e., the U.S. ranked 27 out of 82 countries [85]), medical professionals should consider exploring more practical, less risky options for disordered gamers to acquire financial stability if they report significant monetary involvement in video games. A treatment approach such as this might potentially decrease the likelihood of future relapses and addiction substitution occurring in pathological gamers, which could further promote meaningful change in clients’ lives and inevitably, lead to more successful treatment outcomes [86].

#### 4.4.3. … for Research and U.S. Policy

Few studies, especially in the U.S., have investigated the financial motives and risks that are involved in today’s video games. Therefore, this research is at the forefront in attempting to discern the greater consequences of inserting opportunities for real money, gambling-related or not, into video games available to every child, teenager, and adult with a mobile device. This situation is especially concerning due to the lack of oversight involved in the video game industry’s implementation of gambling features [87,88,89]. Unlike traditional casinos, the gaming industry is allowed to largely self-regulate within the U.S., which is a regulatory setup that may not properly ensure consumers are protected from predatory monetization practices within video games [27].

### 4.5. Limitations

Overall, there were several limitations of this study that warrant acknowledgment. First, this research relied on self-reported cross-sectional data for all analyses, preventing our findings from fully addressing questions of causality between variables. Second, without a diagnostic determination being completed by a medical professional, we can only relay the number of IGD symptoms reported by participants and suggest whether there is a possibility of a medical diagnosis. Third, participants were recruited through convenience sampling to participate in the study’s survey and were not randomly sampled from the U.S. general population, which hinders us from identifying the true response rate of the survey and making more confident generalizations to the larger population. Although it is important to note that the survey’s recruitment description did not indicate the study was examining video gaming. Fourth, MTurk participants in Sample 2 could have been motivated to participate in the study for financial reward. Despite the study’s monetary compensation for MTurk workers being relatively low: USD 2.00, it is still possible that this payment incentivized participation in the survey that would have otherwise not occurred. Fifth, each question within the study’s survey required a response in order to minimize missing values in the datasets. This type of survey design could have prevented the identification of problematic questions or measures.

### 4.6. Future Directions

Future studies should investigate problematic video game behaviors in the U.S. adult population longitudinally to better understand the directionality of associations between video game motives, different types of microtransactions, and IGD symptoms. It would also be beneficial to detail the course of disordered gaming over a significant temporal period to observe how different levels of involvement can occur at certain times in a player’s life. Additionally, the connection between disordered gaming and adverse childhood experiences (ACEs) has not yet been well documented. Previous research on childhood adversity and its relationship to later substance abuse has shown that for each ACE someone reports, it can double or even quadruple their likelihood of early substance abuse [90]. Based on this pattern, as well as the emerging evidence on this topic for gaming disorders [91], it seems highly plausible that a better understanding of these early life experiences might further help the general public and medical professionals identify why problem gaming behaviors develop in some individuals and not others. 

On a related note, there appears to be a need for more neurobiological research on individuals with symptoms of IGD, as well as other behavior-based addictive disorders. While this study focused primarily on the psychosocial aspects of these variables, there is also consistent evidence demonstrating that a person’s biology plays a critical interactive role in the overall expression of addictive behaviors [92]. Future research could further explore the neurobiological pathways related to reward deficiency syndrome, since there is now considerable evidence that suggests this particular syndrome may underlie an array of physical and mental health problems including, but not limited to, addictive disorders [93]. 

## 5. Conclusions


*The question is never “Why the addiction?” but “Why the pain?”*
—Gabor Maté [94] (p. 36)

The prevalence of addiction we are seeing today, perhaps, speaks to a larger issue of how we have structured our modern society for the average person. More than ever before, we appear to be disconnecting from each other, as well as ourselves, which could at least explain some of the reasons why addiction appears to be finding its way into the lives of so many around us. Yet, until we decide to confront the fundamental components of our human culture as it now manifests, maybe the only option is to reduce the harms that can arise from the countless ways people cope with this current era. For video games, we could start by implementing more stringent age verification for games with unlimited spending limits or create greater transparency into the video gaming industry’s use of gambling mechanics in order to reduce the possibility of systematic consumer manipulation for corporate profits. Regardless of the path we pursue, though, it is becoming increasingly apparent that a radical shift is needed to address the existential threat that our contemporary ways of living have created for life on this planet.

On a final note, if economic and social inequalities continue to widen as they have in recent decades [95], here is an issue to ponder: what are the global implications for mental health when the symptoms of numerous disorders stem directly, and indirectly, from the almighty dollar? Our culture in the U.S., for the most part, appears to generally endorse compulsive and indulgent behaviors (e.g., gambling, drinking, shopping, drug use [power included], and now apparently gaming). That is, if you can afford it. Prompting an additional question: as a society, have we legitimized addiction for only the rich and privileged? After all, individuals with a sizeable financial cushion can often avoid the various economic, social, and personal costs (e.g., depression, shame) that frequently arise from addictive behaviors [96], whereas people from socially marginalized groups typically feel the full brunt of the downfall through societal condemnation and institutional punishment. As the wealth gap around the world worsens and monetary barriers to treatment multiply, it is regrettable to know that without drastic changes taking place, one’s social and economic status in this world will only continue to become more predictive of the quality of life that person is able to achieve.

## Figures and Tables

**Figure 1 ijerph-19-15384-f001:**
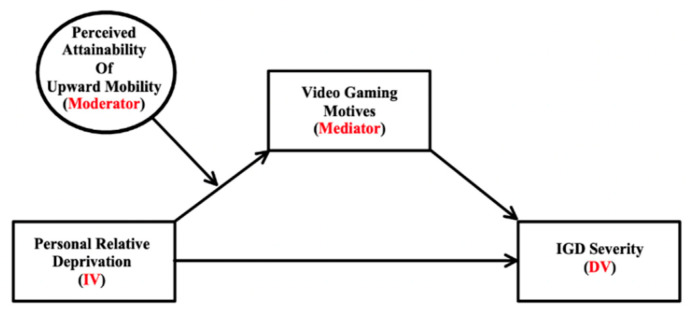
This conceptual diagram displays personal relative deprivation predicting IGD severity via gaming motives (i.e., coping, financial, mood enhancement, and social) at different levels of perceived upward mobility. IV = Independent variable; DV = Dependent variable; IGD = Internet gaming disorder.

**Figure 2 ijerph-19-15384-f002:**
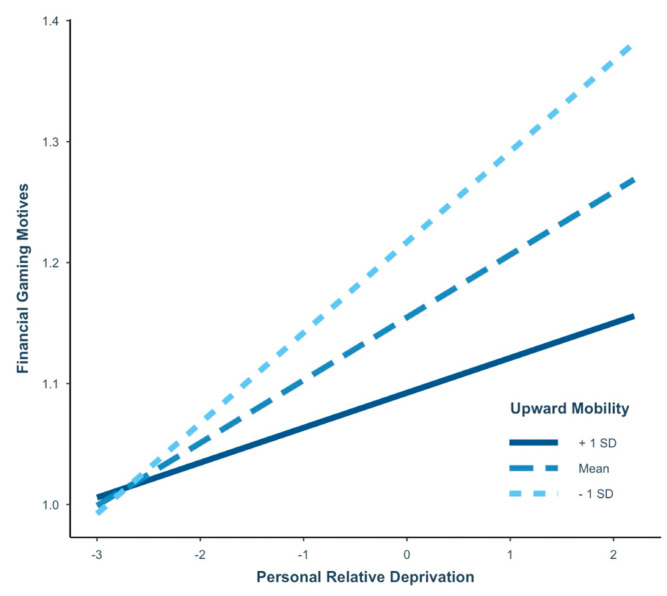
The conditional effect of perceived upward mobility on personal relative deprivation and financial video game motives for Study 1. All variables were coded to reflect more of the construct, with average scores computed for each participant. This figure represents the association between personal relative deprivation and financial gaming motives for U.S. college students at three levels of perceived upward mobility: (1) +1 *SD* above the mean, (2) the mean, and (3) -1 *SD* below the mean. Covariates included gender, income, loneliness, negative urgency, and positive urgency. *n* = 797.

**Figure 3 ijerph-19-15384-f003:**
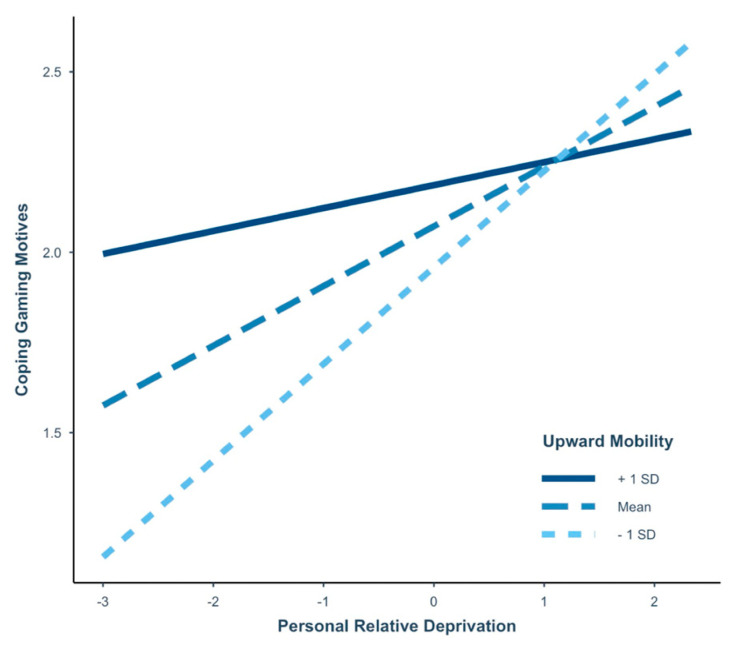
The conditional effect of perceived upward mobility on personal relative deprivation and coping video game motives for Study 2. All variables were coded to reflect more of the construct, with average scores computed for each participant. This figure represents the association between personal relative deprivation and coping gaming motives for MTurk participants at three levels of perceived upward mobility: (1) +1 *SD* above the mean, (2) the mean, and (3) -1 *SD* below the mean. Covariates included gender, income, loneliness, negative urgency, and positive urgency. *n* = 179.

**Table 1 ijerph-19-15384-t001:** Demographic characteristics and video game frequencies for Study 1 and Study 2.

Variable	Study 1 *n* = 797	Study 2 *n* = 179
Male	37.1%	51.4%
Ethnicity/Race		
Asian American	32.1%	9.5%
African American/Black	12.5%	10.6%
Caucasian/White	40.7%	79.9%
Latin or Hispanic American	32%	3.9%
“Other”	13.6%	2.9%
Personal income		
Less than USD 20,000	82.9%	17.3%
USD 20,000–USD 39,999	11.3%	30.7%
USD 40,000–USD 59,999	3.4%	20.1%
USD 60,000 or more	2.4%	31.9%
Gaming frequency (past year)		
2–11 times	33%	12.3%
Monthly	16.9%	16.2%
Weekly or Daily	50%	71.5%
Purchased microtransactions		
Past year	39.1%	28.5%
Spent ≥ USD 100–past month ^1^	12.8%	9.3%
Spent ≥ USD 200, 1 game–lifetime ^2^	18.9%	13.7%
Spent ≥ USD 1000, 1 game–lifetime ^2^	4.6%	2%

Note. A microtransaction, in this context, is an in-game purchase within an online video game. For ethnicity/race variable, percentages exceed 100% because more than one option could be selected. ^1^ Percentages calculated out of participants who reported USD 1.00 or more spent on microtransactions during the most recent month: *n* = 227 for Study 1 and *n* = 43 for Study 2. ^2^ Percentages calculated out of participants who reported USD 1.00 or more spent on microtransactions during their lifetime: *n* = 307 for Study 1 and *n* = 51 for Study 2.

**Table 2 ijerph-19-15384-t002:** Study 1’s descriptive statistics, internal consistencies, and correlations for main variables.

Variable	*M*	*SD*	1	2	3	4	5	6	7	8	9
1. PRD	−0.67	0.91	–								
2. Upward Mobility	0.94	1.05	**−0.30**								
3. Financial Motives	1.13	0.39	**0.13**	**−0.15**							
4. Coping Motives	1.96	0.86	**0.09**	**−0.08**	**0.25**						
5. IGD Severity	1.73	1.91	**0.08**	−0.06	**0.26**	**0.54**					
6. Gender	–	–	**−0.12**	0.02	**0.13**	**0.17**	**0.11**				
7. Loneliness	2.70	0.81	**0.23**	**−0.11**	**−0.11**	**0.09**	**0.07**	−0.01			
8. Negative Urgency	2.23	0.74	**0.22**	**−0.12**	0.06	**0.24**	**0.21**	−0.07	**0.29**		
9. Positive Urgency	1.90	0.73	**0.18**	**−0.12**	**0.19**	**0.26**	**0.26**	**0.13**	**0.19**	**0.60**	
Cronbach’s α	–	–	0.86	0.79	0.79	0.89	0.75	–	0.91	0.76	0.83

Note. IGD = Internet Gaming Disorder; PRD = Personal relative deprivation. Bold values indicate statistical significance. All variables besides gender were coded to reflect more of the construct; gender was coded 0 for females and 1 for males; all motives refer to video gaming. *n* = 797.

**Table 3 ijerph-19-15384-t003:** Study 2′s descriptive statistics, internal consistencies, and correlations for main variables.

Variable	*M*	*SD*	1	2	3	4	5	6	7	8	9
1. PRD	−0.73	1.21	–								
2. Upward Mobility	0.91	1.05	**−0.51**								
3. Financial Motives	1.15	0.41	0.05	**−0.15**							
4. Coping Motives	2.02	0.75	**0.29**	−0.12	**0.15**						
5. IGD Severity	1.15	1.64	**0.22**	−0.10	**0.15**	**0.39**					
6. Gender	–	–	0.07	−0.01	−0.02	−0.14	−0.02				
7. Loneliness	2.13	0.90	**0.61**	**−0.42**	−0.01	**0.27**	**0.31**	0.06			
8. Negative Urgency	1.51	0.58	**0.17**	**−0.17**	**0.33**	**0.24**	**0.28**	0.00	0.14		
9. Positive Urgency	1.83	0.72	**0.37**	**−0.22**	**0.20**	**0.29**	**0.41**	**−0.01**	**0.42**	**0.70**	
Cronbach’s α	–	–	0.93	0.84	0.87	0.86	0.76	–	0.96	0.84	0.80

Note. IGD = Internet Gaming Disorder; PRD = Personal relative deprivation. Bold values indicate statistical significance. All variables besides gender were coded to reflect more of the construct; gender was coded 0 for females and 1 for males; all motives refer to video gaming. *n* = 179.

**Table 4 ijerph-19-15384-t004:** Moderated mediation summary for Study 1.

Predictors	Outcome: Financial Motives	
	*B*	95% CI*_B_*
Personal relative deprivation (PRD)	**0.07 *****	**0.03, 0.11**
Perceived upward mobility (UM)	−**0.06 *****	−**0.09,** −**0.03**
PRD × UM	−0.02	−0.05, 0.002
Effect of PRD at low UM (−1 SD)	**0.07 *****	**0.03, 0.12**
Effect of PRD at average UM	**0.05 *****	**0.02, 0.08**
Effect of PRD at high UM (+1 SD)	0.03	−0.01, 0.07
Model summary: ***F*(8788) = 11.80 *****	*R*^2^ = 10.7%	
**Predictors**	**Outcome: IGD Severity**	
	* **B** *	**95% CI*_B_***
PRD	−0.003	−0.15, 0.14
Financial gaming motives	**1.07 *****	**0.73, 1.41**
Conditional indirect effect		
Low UM	**0.08**	**0.04, 0.14**
Average UM	**0.06**	**0.02, 0.10**
High UM	0.03	−0.002, 0.07

Note. All variables were coded to reflect more of the construct. Bold values indicate statistical significance, which was determined using 95% bias-corrected bootstrapped confidence intervals that did not include zero (based on 5000 bootstrapped samples). This model displays unstandardized coefficients for personal relative deprivation predicting IGD severity via financial gaming motives at three levels of perceived upward mobility. Conditional indirect effects were not assigned a *p*-value in PROCESS macro 3.4 output [79]. Covariates included gender, income, loneliness, negative urgency, and positive urgency. *** *p* < 0.001; *n* = 797.

**Table 5 ijerph-19-15384-t005:** Moderated mediation summary for Study 2.

Predictors	Outcome: Coping Motives	
	*B*	95% CI*_B_*
Personal relative deprivation (PRD)	**0.26 *****	**0.12, 0.40**
Perceived upward mobility (UM)	0.09	−0.01,0.19
PRD × UM	**−0.08 ***	**−0.15, −0.01**
Effect of PRD at low UM (−1 SD)	**0.27 *****	**0.12, 0.41**
Effect of PRD at average UM	**0.17 ****	**0.05, 0.28**
Effect of PRD at high UM (+1 SD)	0.06	−0.08, 0.21
Model summary: ***F*(7171) = 6.33 *****	*R*^2^ = 20.6%	
**Predictors**	**Outcome: IGD Severity**	
	* **B** *	**95% CI*_B_***
PRD	−0.10	−0.33, 0.12
Coping gaming motives	**0.64 *****	**0.33, 0.94**
Conditional indirect effect		
Low UM	**0.17**	**0.06, 0.32**
Average UM	**0.11**	**0.01, 0.22**
High UM	0.04	−0.07, 0.16

Note. All variables were coded to reflect more of the construct. Bold values indicate statistical significance, which was determined using 95% bias-corrected bootstrapped confidence intervals that did not include zero (based on 5000 bootstrapped samples). This model displays unstandardized coefficients for personal relative deprivation predicting IGD severity via coping gaming motives at three levels of perceived upward mobility. Conditional indirect effects were not assigned a *p*-value in PROCESS macro 3.4 output [79]. Covariates included gender, income, loneliness, negative urgency, and positive urgency. * *p* < 0.05; ** *p* < 0.01; *** *p* < 0.001; *n* = 179.

## Data Availability

Data for this research are available upon request; please contact the first author for details.

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
