# Peer review of "Do Gamers Play for Money? A Moderated Mediation of Gaming Motives, Relative Deprivation, and Upward Mobility"

_ijerph, 2022, doi:10.3390/ijerph192215384_

Round 1
Reviewer 1 Report
This is a very well-written paper with important findings. I recommend acceptance but would like to a make a few suggestions. The authors of this study seem to be involved with psychological aspects not molecular neurobiology and even genetics. With that stated some of their remarks remis the neurogenetic and neuro-epigenetic mechanisms linked to IGD etc. Therefore, I suggest that they at least make mention of "Reward Deficiency Syndrome" as related to all addictive behaviors both substance and non-substance. (see Blum K, McLaughlin T, Bowirrat A, Modestino EJ, Baron D, Gomez LL, Ceccanti M, Braverman ER, Thanos PK, Cadet JL, Elman I, Badgaiyan RD, Jalali R, Green R, Simpatico TA, Gupta A, Gold MS. Reward Deficiency Syndrome (RDS) Surprisingly Is Evolutionary and Found Everywhere: Is It "Blowin' in the Wind"? J Pers Med. 2022 Feb 21;12(2):321. doi: 10.3390/jpm12020321. PMID: 35207809; PMCID: PMC8875142.)
Also the subjects are young adults and we now know young adults have issues with decision making based on unmyelinated Pre-frontal cortex (seeBlum K, Febo M, Smith DE, Roy AK 3rd, Demetrovics Z, Cronjé FJ, Femino J, Agan G, Fratantonio JL, Pandey SC, Badgaiyan RD, Gold MS. Neurogenetic and epigenetic correlates of adolescent predisposition to and risk for addictive behaviors as a function of prefrontal cortex dysregulation. J Child Adolesc Psychopharmacol. 2015 May;25(4):286-92. doi: 10.1089/cap.2014.0146. Epub 2015 Apr 28. PMID: 25919973; PMCID: PMC4442554.)
I would like to see a small section related to these above remarks before final acceptance
Author Response
To the Reviewers:
Thank you for your compliments and feedback on our paper! We are both extremely appreciative to have the opportunity to be a part of this special issue.
- Reviewer #1: Regarding your suggestions, we have added a new paragraph in our Future Directions section that cites both of the mentioned studies and their applicability to this research; we agree that biological factors are a crucial part of addictive disorders, so we are glad that you pointed that out for us to mention; the new paragraph is currently highlighted in yellow on the revised draft to reflect those changes.
Note: The same content is within the attached Word document.

Reviewer 2 Report
Dear Authors,
Congratulations on conducting these very pertinent, interesting and methodologically sound studies on relative deprivation and upward mobility and their relationship with gaming and gambling. Your studies provide strong evidence of the potential health and social risks associated with video gaming, and insight into the role of PRD, UM and coping motives. This evidence is extremely useful to the emergent field of video gaming research, as well as to gambling research.
Your paper was very engaging and interesting to read. You have presented a very good review of the relevant literature, and effectively presented the rationale for your study. Your hypotheses are appropriate and well articulated.
PRD and UM are challenging factors to operationalise and reliably measure. I believe that you have successfully achieved that for the purposes of your study. Your sampling methods are good, and you appropriately report their limitations. You have also used standardised measures where possible. You used a retrospective self-report design, which is a limitation to the strength of the evidence, and you acknowledge this.
Your analyses are appropriate for your hypotheses and are reported with clarity and insight. I could only suggest that the figure of your model be reproduced, showing the regression coefficients. I find these faster to examine and understand than tables. Of course, this is only a personal preference.
Your discussion demonstrates a very good understanding of the literature and the implications of your findings from theoretical, practice and policy perspectives. I appreciate the insightful discussion you presented in your conclusion.
Your paper has left me without much to say as a reviewer, other than to say congratulations and I hope that your paper is widely read and influential in the further development of video game and gambling research.
Kind regards,
Reviewer
Author Response
To the Reviewers:
Thank you for your compliments and feedback on our paper! We are both extremely appreciative to have the opportunity to be a part of this special issue.
- Reviewer #2 Comment: "I could only suggest that the figure of your model be reproduced, showing the regression coefficients."
- Thank you for the suggestion; we did consider this approach and do agree that it is typically easier to understand the regression coefficients in a figure than in a table; however, since we are displaying three levels of the moderator and three levels of the indirect effect for each model, we settled on using tables for our results in order to avoid making figures that may have appeared cluttered to some readers; we did adjust the previous row alignments on our tables, though, to hopefully make them easier to skim.
Note: The same content is contained within the attached Word document.
